# Classification of flavors in cigarillos and little cigars and their variable cellular and acellular oxidative and cytotoxic responses

Gina R. Lawyer☯, Monica Jackson☯, Melanie Prinz, Thomas Lamb, Qixin Wang, Thivanka Muthumalage, Irfan Rahman[ID]*

Department of Environmental Medicine, School of Medicine & Dentistry, University of Rochester Medical Center, Rochester, NY, United States of America

☯ These authors contributed equally to this work.
* Irfan_rahman@urmc.rochester.edu

**Data Availability Statement:** All relevant data are within the manuscript and its Supporting

## Abstract

Flavored tobacco products are increasing in popularity but remain unregulated, with the exception of the ban on flavored conventional cigarettes. Lack of regulation of cigarillos and little cigars allows vendors to have their own version of popular flavors, each with different chemical components. A new classification system was created for flavored cigars in order to easily communicate our results with the scientific community. To understand the physico-chemical characteristics of flavored little cigars and cigarillo smoke, size distribution and concentration of particulate matter in smoke were determined. Acellular reactive oxygen species generation was measured as an indirect measurement of the potential to cause oxidative stress in cells. In addition, cigarillo smoke extract treatment on bronchial epithelial (Beas-2b) cells were assessed to determine the flavor-induced cellular toxicity. Flavored cigars/cigarillos showed significant variability in the tested parameters between flavors as well as brands of the same flavor, but most of the cigars showed higher potential of generating oxidative stress, than research grade cigarettes. Flavored cigars produced maximum particle concentrations at 1.0μm and 4.0 μm compared with 3R4F reference research cigarettes. A differential cytotoxic response was observed with cigarillo smoke extract treatments, with "fruits/candy" and "drinks" being the most toxic, but were not more cytotoxic than smoke from cigarettes. These cigarillos with flavors must be well characterized for toxicity in order to prevent adverse effects caused by exposure to flavor chemicals. Our study provides insight into understanding the potential health effects of flavor-infused cigars/cigarillos and the need for the regulation of those flavoring chemicals in these products. Future research is directed to determine the flavoring toxicity of little cigars and cigarillos in vivo studies.

## Introduction

In the current environment where the focus is on the dangers of cigarettes, cigars are often perceived to be safer than cigarettes due to the lack of public attention and scientific knowledge

Information files. Please see Table 1 and Figs 1–6, and Supporting info S1 Table.

**Funding:** This work was supported in part by a National Institutes of Health (NIH) Grants, NIH 1R01HL135613, NIH 1R01HL085613, and the Food and Drug Administration (FDA) Center for Tobacco Products (CTP). Also in part was supported by the National Cancer Institute of the National Institutes of Health (NIH) and the Food and Drug Administration (FDA) Center for Tobacco Products under Award Number U54CA228110. The funders had no role in study design, data collection and analysis, decision to publish, or preparation of the manuscript.

**Competing interests:** The authors have declared that no competing interests exist.

[1–3]. From the year 2000–2011, there was a 33% decrease in consumption in cigarettes and a 123% increase in the use of non-cigarette products, like cigars [4]. This change in behavior is also seen in youths where it was reported in 2012 that cigars were the second most common way for youths to use tobacco [5]. Smoking a cigar carries the same health risks as smoking a cigarette [6, 7].

The amount of legislation pertaining to cigars is much lower when compared to other tobacco products. In 2009, flavored cigarettes were banned by the Family Smoking Prevention and Tobacco Control Act; however, there is no such legislation for cigar products [5]. In association with this ban, there was a significant increase in cigar use seen in high school students, especially in flavored cigars/cigarillos [8]. Adding flavors reduces the harshness of tobacco, making flavored cigars an easy gateway to youth addiction and new tobacco user recruitment [1, 9]. These flavors are not just found within the filler of the cigar, but also extra sweetener is seen the wrappers and mouth tips that make direct contact with saliva [10]. Young adults, 18–24 years old, have the highest usage of flavored cigar products [5].

There is very little information available on what makes up these flavoring chemicals, how the flavor is added, and how these flavors differ by brand [11]. This can be attributed to flavors being a mixture of many compounds, where there is no universal combination method to create more complex tastes [11]. The same flavoring chemicals found in food and drink products are also seen in these flavored cigars [12]. Inhalation of a common flavoring chemical, diacetyl, is linked to the irreversible damage in the lung tissue causing "popcorn lung disease" or bronchiolitis obliterans[13].

A quantitative way of measuring oxidative stress-mediated toxicity is by determining the concentration of reactive oxygen species (ROS) produced by the flavored cigars [14]. ROS are cellular byproducts of the electron transport chain in the mitochondrial membrane and intracellular NADPH oxidase machinery [15]. In addition, ROS can be formed extracellularly through reactions among various radicals, including hydroxyl radicals. Oxidative stress is damaging to cellular components and biomolecules by causing DNA damage, inhibition of apoptosis, and activation of proto-oncogenes [16], leading to the pathogenesis of COPD and lung cancer [17].

Particulate matter (PM) released by cigars may contribute to tissue destruction of the lungs. When the cigar is lit, a combustion reaction occurs and produces vapor to be inhaled that contains PM, for instance:mainstream cigarette smoke contains 10,000–40,000 μg of PM [18]. PMs with respirable size, such as PM 2.5 and PM 1.0, are capable of depositing in the lungs, and can even be trapped in the alveolar region, thereby leading to damage of the lung tissue [18]. Further, extensive research has been conducted on mainstream cigarette smoke, but there is a lack of understating in flavored cigar/cigarillo smoke constituents.

Currently, there is no standard nomenclature for the classification of cigar flavors; this has become a challenge for communication among cigar users and in the scientific community [19]. This lack of organization does not allow for a definitive way to compare work from separate labs, and classify a vast and different cigar flavors available currently.

We hypothesize that the similarities in toxicity between similarly flavored cigars or even within a brand will be able to categorize which cigars are the most deleterious. This knowledge will help to alert not only flavored cigar smokers, but also to make a broader assumption to which flavoring chemicals, in general, are the most harmful. The objective of this study was to determine if the chemical composition in flavored cigars produces differential oxidative and cytotoxic responses. We tested the hypothesis by using various commercially available flavored little cigars and cigarillos in acellular and cellular systems.

## Materials and methods

### Ethics statements

All experiments performed in this study were approved and in accordance with the University of Rochester Institutional Biosafety Committee. And, all protocol, procedure and data analysis in this study were followed the NIH guildlines and standards of reproducibility and scientific rigor by an unbiased approach.

   Animal or Human study protocol: None
   Institutional biosafety approvals: Yes

### Scientific rigor statement

The approach to creating the experiment was unbiased and analysis done on the results ensured that our data are reproducible.

### Cigar procurment and categorization

Little cigars and cigarillos were purchased in Rochester, NY, at various locations and vendors, to be tested in this study (see S1 Table). The types of cigars used in the experiment are classified into six basic categories based on names and descriptions on the packaging. Flavor names are generated by the manufacturer and a visual representation of the flavors can be found in Fig 1. Categories used are tobacco, menthol, fruit, candy, drinks, and spices. These categories

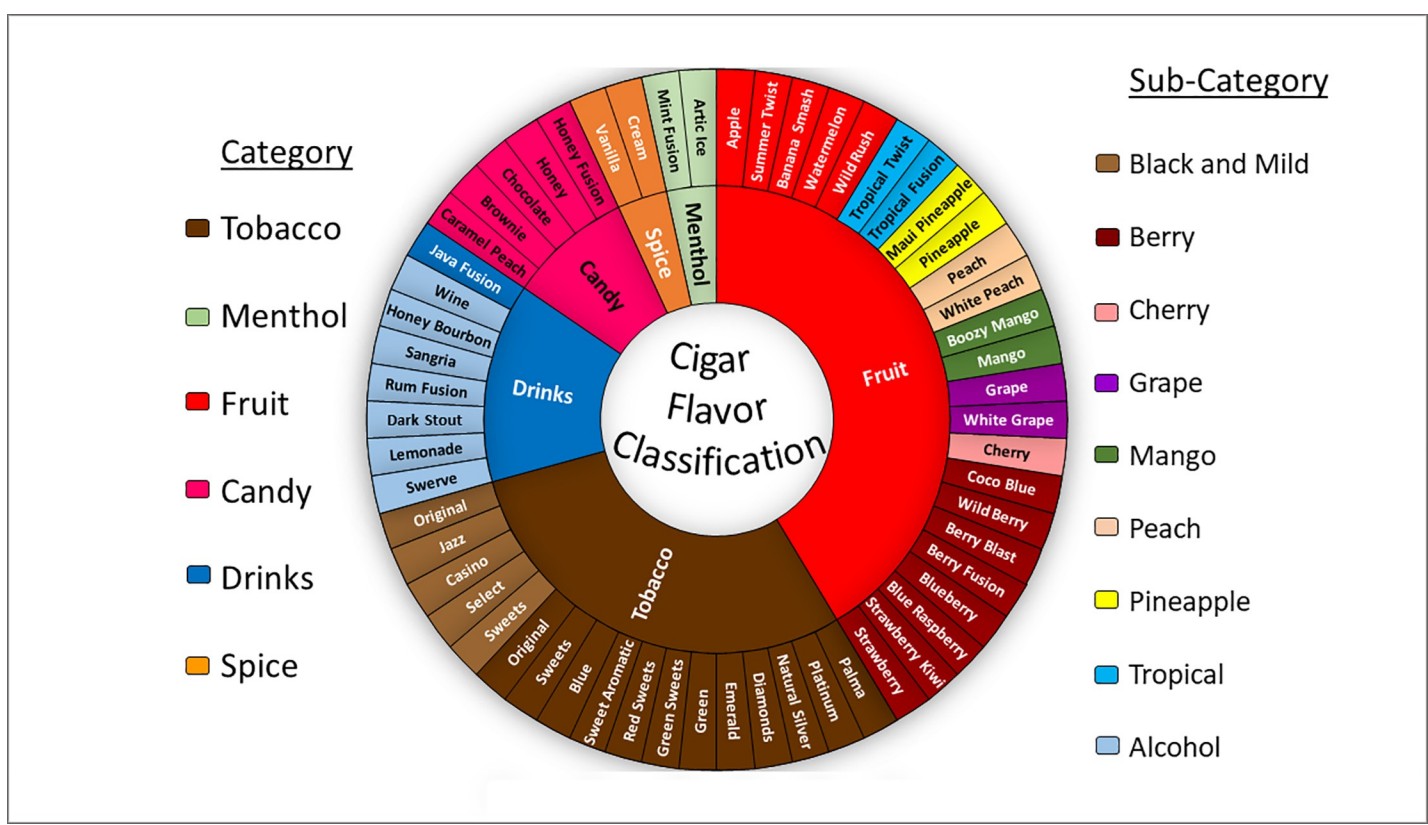

**Fig 1. Classificaton of cigar flavors.** The flavors listed are based on products available on the market but is not a fully inclusive list. The flavors are organized in a new classification system. The inner circle is shaded based on the primary flavor category of the cigar. The outer circle correspond to the sub-category of flavors within the primary category.

were further sub-divided as needed which included, Black and Mild, berry, cherry, grape, mango, peach, pineapple, tropical, and alcohol. Some variation in which flavored cigar was used is present between each expirmental test due to the limitations in inventory.

## Cell-free ROS assay

A fluorogenic dye was created using 0.01N NaOH, 2'7' dichlorofluorescein diacetate (H$_2$DCF-DA,) (EMD Biosciences, CA) (Cat # 287810), PO$_4$ buffer made of sodium phosphate monobasic (JT Baker, NJ) (Cat #3828–01) and sodium phosphate dibasic (Sigma- Aldrich, MO) (Cat #2–0751), and horseradish peroxide (HRP) (Thermo Fisher, Ma) (Cat # 31491). ROS were detected based on flouresent intensity at 495/529nm. Standards that ranged from 0 to 50 µM were created using a 1mM hydrogen peroxide stock that was reacted with the fluorogenic dye at 37˚C for 15 minutes. Standards were measured on a spectrofluorometer (Turner Quantech fluorometer, Mo. FM109535) in fluorescence intensity units (FIU). Samples of cigar smoke extract were also given 15 minutes at 37˚C to react with the fluorogenic dye, then measured immediately. Sample readings were based on the hydrogen peroxide standard curve and denoted as "hydrogen peroxide H$_2$O$_2$ equivalents".

Three different methods were used for creating cigarillo smoke extract using an impinger mechanism. Multiple methods were tested to fully understand the cigar burning process. Method 1 used an SKC lab pump (model 224-PCXR8, Eighty-Four, PA, USA) with an average flow rate of 1 L/min. Little cigars or cigarillos were attached to a 50 mL conical tube containing 10 mL of freshly made fluorogenic dye. Cigarillos were lit and smoke was bubbled through the dye for 1 minute. A new cigarillo was used for each sample. Immediately following the bubbling of the dye, it was filtered through a sterile 0.45 µm polyethersulfone syringe filter to eliminate any debris. Method 2 used a standard lab vacuum to bubble the cigarillo smoke. A cigarillo was attached to a 50 mL conical tube containing 10 mL of freshly made fluorogenic dye and smoke was bubbled for 10 seconds. This process was repeated two to three times on the same cigarillo to create separate samples of extract. Immediately following the bubbling of the dye, it was filtered through a sterile 0.45 µm polyethersulfone syringe filter to eliminate any debris. Method 3 used a standard lab vacuum to bubble the cigarillo smoke. Each cigarillo was sectioned into three equal pieces by weight and attached to a 50 mL conical tube containing 10 mL of freshly made fluorogenic dye. Cigar smoke was bubbled through the dye at a constant rate as described above. Each portion of the split cigarillo was bubbled until completely burned and measured for ROS in H$_2$O$_2$ equivalents.

## Cell culture and treatment with little cigar/cigarillo smoke extract (CSE)

Human bronchial epithelial cells (Beas-2b) were obtained from ATCC, USA. Cells were cultured and grown at 37˚C in Corning® DMEM (Dulbecco's Modified Eagle's Medium) /Hams F-12 50/50 mix media (Cat # 16-405-CV) with added 5% fetal bovine serum (FBS), 1% penicillin streptavidin (P/S) (Corning, NY) (Cat # 30-002-Cl), and 15 mM HEPES (Alfa Aesar, MA) (Cat #J60712). At 85–90% confluency, cells were serum-deprived with DMEM media with added 1% FBS, 1% P/S, and 15 mM HEPES for 8 hours.

Beas-2b cells were seeded in six-well plates at a density of 400,000 cells per well. Cells were grown until 85–90% confluency and then serum-deprived for 8 hours. Cells were treated with 0.5% and 1.0% concentrations of CSE. The smoke extract was created by using method #1; however, in place of fluorogenic dye being bubbled, 10 mL of 1x PBS was used. The extract was then measured on a Beckman Coulter spectrophotometer (Model DU520). An absorbance value of 1.00 ± .05 was considered to be 10% concentration, and further dilutions were done to

obtain that concentration. Cells were treated for twenty-four hours. Untreated cells remained in DMEM supplemented with 1% FBS.

The viability of the Beas-2b cells was measured using acridine orange (AO) and propidium iodide (PI) staining. 20 μL of AO/PI staining and 20 μL of live cells were combined, and then 20 μL of the mixture was inserted into the Nexcelom's Cellometer (Model Auto 2000) and analyzed. The analysis included the number and concentration of live, dead, and total cells and the percent viability of the sample.

## Particulate matter collection and its concentration distribution in cigar smoke

To characterize the particle size distribution in cigar smoke, TSI[TM]'s Dust Trak II (model 8530) was used with particle diameter cut-offs at 1 μm, 2.5 μm, 4 μm, and 10 μm at a 2 L/min sampling rate. Cigars were manually lit and puffed at 1.7 L/min for 1 minute using Scireq's Inexpose system, with 2.5s puff duration, and 16.6s inter-puff interval. Cigar smoke particle sizes were measured in an Enzyscreen chamber (Cat. CR1601) with dimensions 22 cm x 14.5 cm x 16 cm and in a DSI[TM] chamber with dimensions 44.8 cm x 30.1 cm x 29.6 cm [20].

## Statistical analysis

Statistical analysis of significance was calculated using one-way ANOVA as well as Tukey's *post-hoc* test for multiple comparisons by GraphPad Prism Software version 8.1.1. The results are shown as mean ± SEM of average n = 3 to 4. Data were considered to be statistically significant for P values <0.05.

# Results

## Categorization of cigars by flavor

To understand how the flavorings in cigars can be grouped, a new classification system was created (**Fig 1**). This design was used to allow for categories and subtypes. Presently, there are no classifications that convey the flavors in a meaningful way. This figure attempts to do that in a manner which is easy to understand and follow for toxicological studies. There are six main flavor categories, drinks, fruit, tobacco, menthol, candy, and spices with sub-categories, alcohol, black and mild, berry, tropical, pineapple, peach, grape, watermelon, mango, and cherry. Fruit category contained the most flavors as well as the largest number of subtypes.

## Acellular ROS production by little cigars and cigarillos

Little cigars and cigarillos produced differential $H_2O_2$ equivalents (**Figs 2–6**). For method #1, many of the flavored cigarillos showed a significant increase in acelluar ROS compared to air with the exeption of Black and Mild's tobacco, Swisher Sweets' blueberry, cherry from both Pt. Rillos and Jackpot, Swisher Sweets' boozy mango, tropical fusion and caramel peach and finally Black and Mild's wine. The highest aceulluar ROS produced was from Dutch Masters' Mint Fusion, Game's white grape and berry blast, White Owl grape, and Wine by Dutch Masters. Little cigars tested from Djarum, black and special, were both significantly higher in ROS than air and produced more ROS than a 3R4F. While each category has at least one low ROS producing cigar many of these cigars tested have higher ROS than a 3R4F (**Fig 2**).

Method # 2, similar to method #1 showed many of the cigars and cigarillos tested having signicantly higher acelluar ROS production than air (**Fig 3**). Tested cigars were grouped based on their categories and depicted in **Fig 3**. Highest significance was seen in the categories Black and Mild, Tobacco, and Spice with both Djarum cigars being the most significant (**Fig 3**).

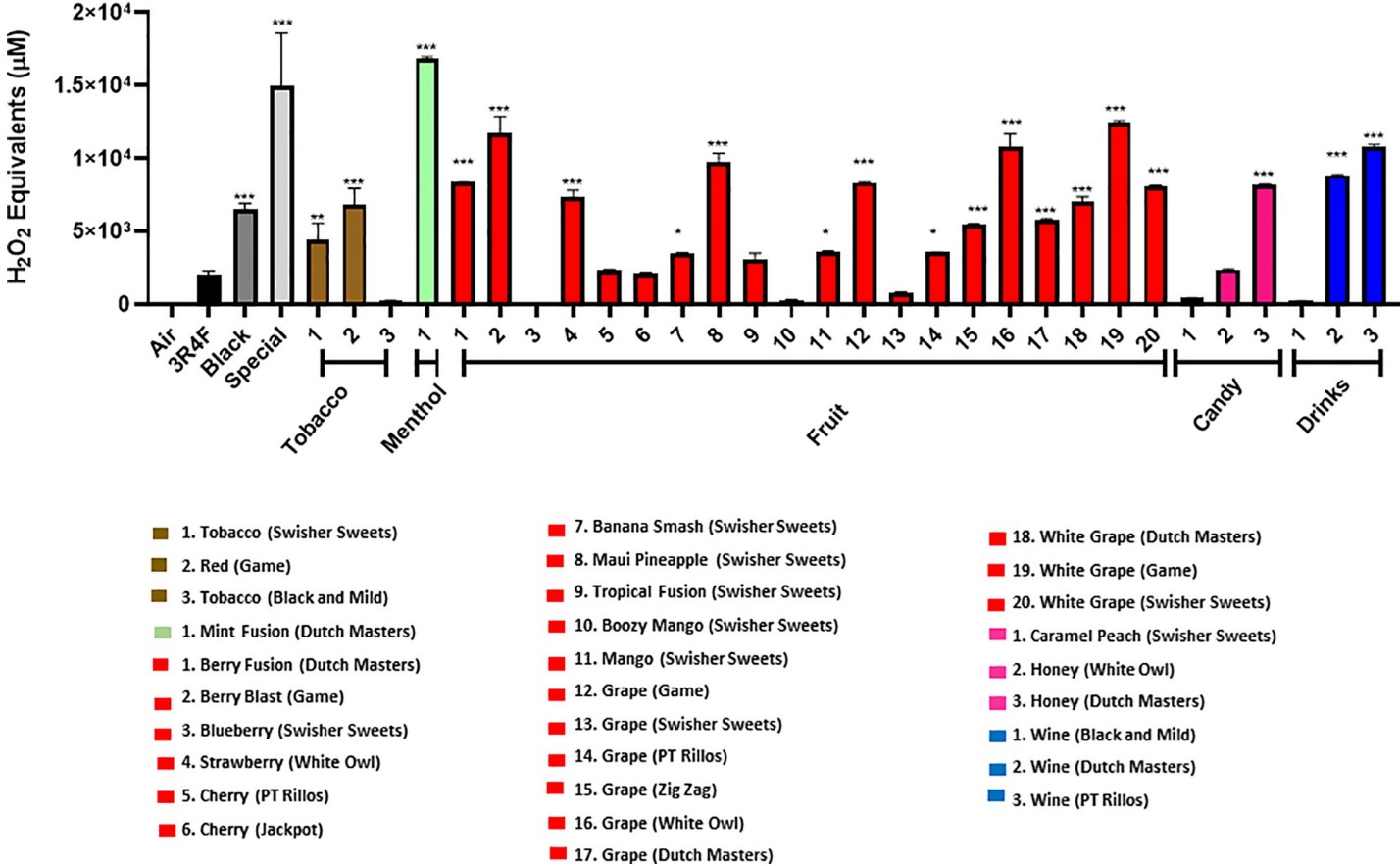

**Fig 2. Generation of ROS by different flavors and brands.** Acellular ROS was measured in samples of various CSE (3R4F and others) using a hydrogen peroxide standard. The CSE was obtained using puffing method #1, where a SKC lab pump was used to continuously puff a cigar or cigarillo for 1 minute at 1 L/min. Names of cigars/cigarillos are listed below with their brand in parenthesis. Combined results of all the flavors tested by method #1. All flavors were compared to the control value of air. Data shown as mean ± SEM, and significance was determined by one-way ANOVA. * $p < 0.05$, ** $p < 0.01$, and *** $p < 0.001$ versus air controls.

Lowest ROS production was seen in categories Candy, Grape, Mango and Peach. Individual cigars and cigarillos tested for ROS productions which show a differential response (**Figs 4 and 5**). Also, we are interested in how different segments of cigars would affect the ROS production. There was no significant difference among all the segment (**Fig 6**) based on method #3 of ROS assay, which means cigar burned in a universal way, and oxidative stress generated continously and universally.

Most notably, between methods #1 and #2, it is easy to see that each method gives significantly different results even when testing the same cigarillo. Within method #1, many of the cigarillos burned down faster than others from the same brand and flavor. This resulted in a wide variations of ROS values for a single cigarillo. For example, PT Rillo's wine flavor was tested four separate times using four separate cigarillos. The range of these tests was 49.6–10,788.80 $H_2O_2$ equivalents.

## Fine particle emission and distribution by flavored little cigars and cigarillos

For all particle size measurements, research cigarettes (3R4F/1R6F) released lower concentration of particles, i.e., particulate matter (PM). The lowest PM from cigars/cigarillos tested was

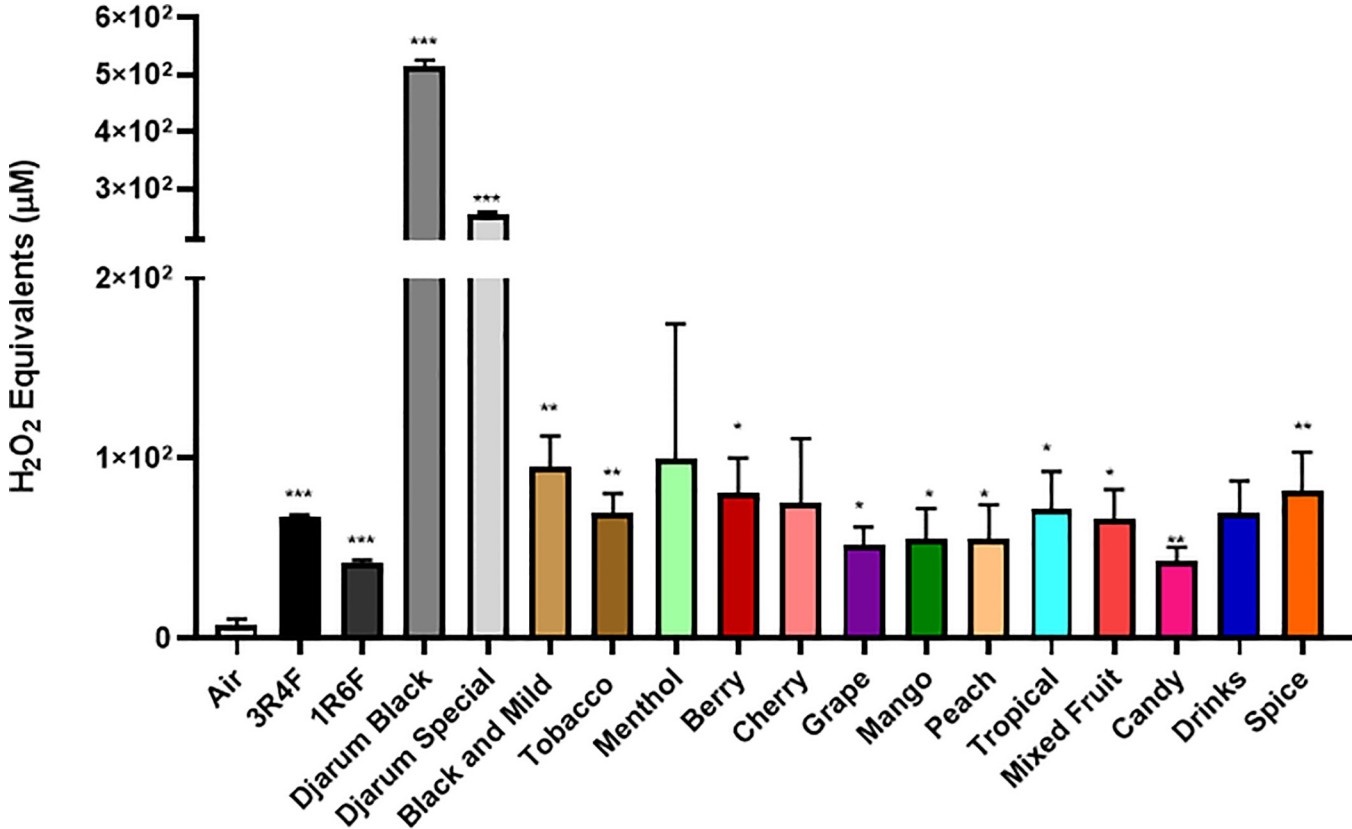

**Fig 3. Generation of ROS by different flavors and brands.** Acellular ROS was measured in samples of various CSE (3R4F and 1R6F) using a hydrogen peroxide standard. The CSE was obtained using puffing method #2, where a general vacuum lab pump was used to continuously puff a cigar or cigarillo for 10 seconds. Names of cigars/cigarillos are listed below with their brand in parenthesis. Combined results of all the flavors tested by method #2. All flavors were compared to the control value of air. (. Data are represented as mean ± SEM, and significance was determined by one- way ANOVA. * $p< 0.05$, ** $p< 0.01$, and *** $p< 0.001$ versus air controls.

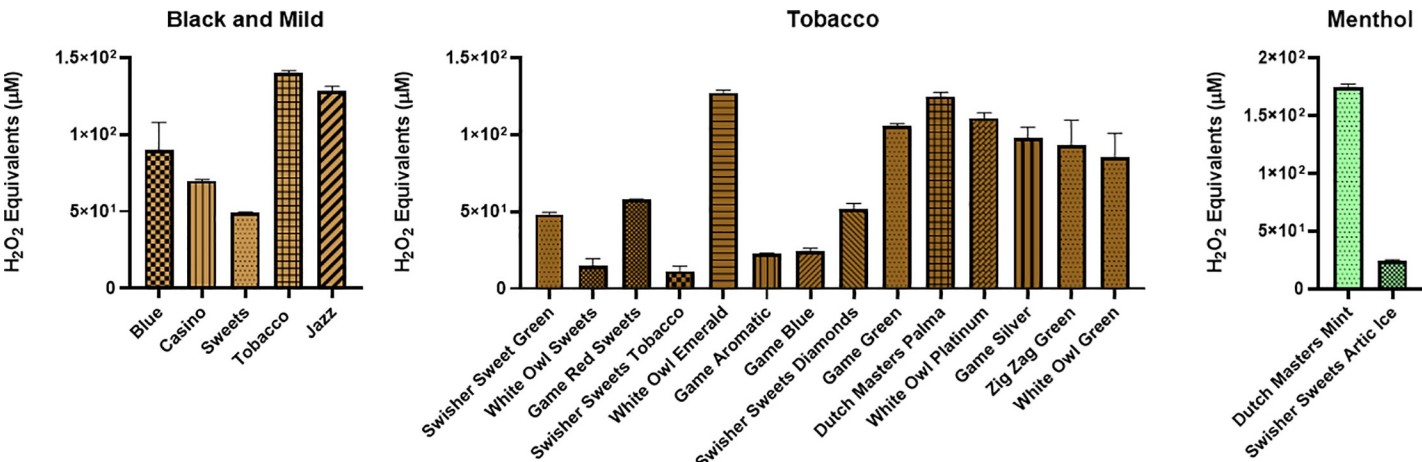

**Fig 4. Generation of ROS by tobacco and menthol flavors and brands.** Acellular ROS was measured in samples of various cigarillos using a hydrogen peroxide standard. The puffing method #2 was used, where a general vacuum lab pump was used to continuously puff a cigar or cigarillo for 10 seconds. Names of cigars/cigarillos are listed below with their brand in parenthesis. Black and Mild, Tobacco, and Menthol flavors were compared to the control value of air. Data are represented as mean ± SEM.

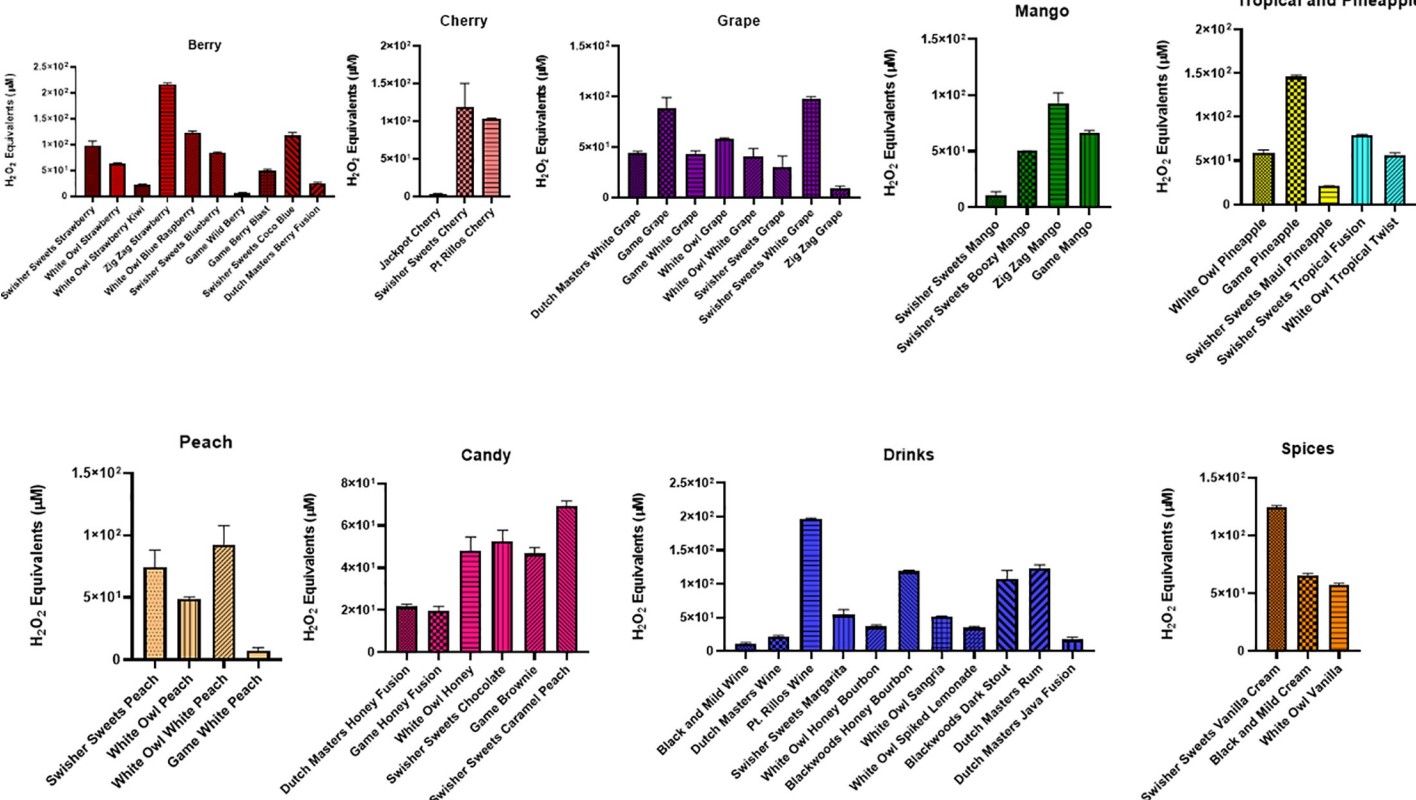

**Fig 5. Generation of ROS by fruit, candy, drink, and spice flavors and brands.** Acellular ROS was measured in samples of various cigarillos using a hydrogen peroxide standard. The puffing method #2 was used, where a general vacuum lab pump was used to continuously puff a cigar or cigarillo for 10 seconds. Names of cigars/cigarillos are listed below with their brand in parenthesis. Fruit, berry/cherry, candy, drinks, and spice flavors were compared to the control value of air. Data are represented as mean ± SEM.

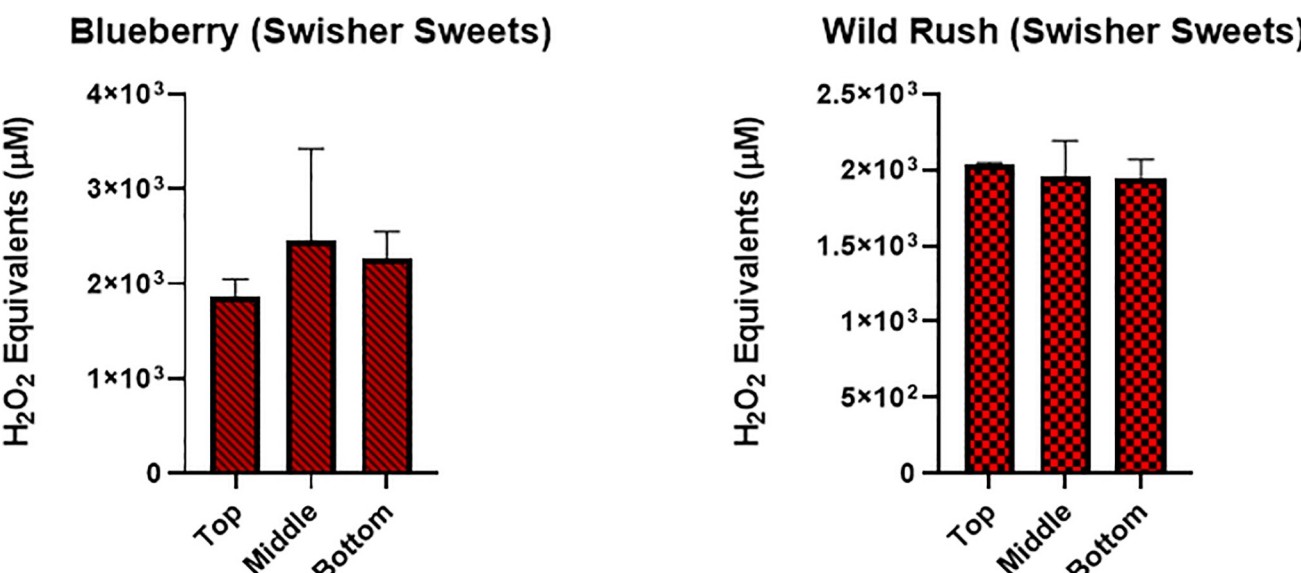

**Fig 6. Generation of ROS by different portions of a cigar.** Acellular ROS was measured in samples of CSE from 2 different cigarillos using a hydrogen peroxide standard. CSE was collected using method #3, where a cigar was partitioned into 3 segments and a general vacuum lab pump was used to continuously pull smoke until cigar was completely burned. Representative example includes Blueberry and Wild Rush cigarillos. Data are shown as mean ± SEM, no significant differences between the sections/segments.

black by Djarum, and the highest was honey fusion by Dutch Masters, special by Djarum, and wine by Black and Mild (Table 1). Most of the cigar samples tested had the highes PM concentration in categories 1 μm and 2.5 μm (Table 1). Fig 7A shows the average PM concentration for each of the seven categories and two reference cigarettes only to be used as a visual representation of Table 1. Fig 7B depicts the PM concentrations of one cigar per category at 1.0 μm. Mint fusion by Dutch Masters and mango by Swisher Sweets have the highest concentrations, whereas a reference cigarette and tobacco by Swisher Sweets are the lowest.

## Exposure to cigar flavorings induces variable cellular toxicity

To determine the cytotoxicity of flavored cigars, Beas-2b cells were treated 0.5% and 1% cigar/cigarillo extract. Overall, each cigar showed a decrease in cell viability from 0.5% treatment to 1% treatment except for Djarum's black cigar (Fig 8). Surprisingly, djarum cigars, tobacco flavor and menthol flavor do not show any significant decreasing in cell viability, while all other fruit, candy, and drink flavors showed reduction of viability. Most significant cytotoxicity can be seen in PT Rillos' wine flavored cigarillo that had viability reduced to 43.9% with 0.5% extract treatment and with a 1% treatment, viability reduced further to 33.6%. At 1% extract treatments Swisher Sweet's caramel peach had a viability reduction to 36.97%, PT Rillos' Strawberry was reduced to 27.2%, and GAME's pineapple reduced to 49.2%. Cell treatment with a 1% extract of a 3R4F cigarette showed a significant reduction in cell viability as well, with an average of 27.7%. Almost all cigars tested had a reduced in cell viability compared with the control, but were not more cytotoxic than a research cigarette (Fig 8).

## Discussion

This study intended to determine the toxicity of flavored cigarillos and little cigars. Selected little cigars/cigarillos from various flavor categories were combusted and acellular ROS, particle concentration/distribution, and the cell viability were assessed. Classification of flavored cigars/cigarillos should provide a convenient nomenclature/vocabulary in the scientific community as there was no scientific classification of cigar flavorings until the present. The classification system created, allows for the categorization of flavors in a way that groups like chemicals together, in an attempt to single out which flavors can be the most toxic. Otherwise, introduction of additional flavorings in cigars or any tobacco products will make comparison between categories more difficult.

The ROS produced by tobacco products are complex and not fully understood. However, it can be broken down into two phases, the gas and the tar phase [21]. Mainstream tobacco smoke constituents react with each other forming reactive oxygen and nitrogen species, including superoxide, hydrogen peroxide, hydroxyl radical, and peroxynitrites [22]. Oxidation of biomolecules, such as proteins and lipids and the formation of DNA lesions are mechanisms of toxicity of tobacco smoke exposure and disease progression [23]. It is widely known that cigarette smoke generates ROS and has significant carcinogenic properties as depicted in dose-dependent studies [24, 25]. In this study, we observe the ROS generated by most of the cigarillos tested in our study is significantly higher than conventional research cigarettes (University of Kentucky 3R4F or 1R6F). This suggests that cigarillos and cigarettes not only induce higher oxidative stresses, but different brands of cigarillos and little cigars showed variable toxicological effects dependent on different chemicals generated during combustion. Corroborating our data, another study showed a greater emission of total semivolative organic compounds (SVOC) and volatile organic compounds (VOC) in cigarillo smoke compared to regular cigar or cigarette smoke [26].

**Table 1. Comparative particulate matter concentration of different particle size in flavored cigarillos comparing with research cigarettes.**

| Category | Name (Brand) | 1.0 μm | 2.5 μm | 4.0 μm | 10.0 μm |
|---|---|---|---|---|---|
| 3R4F | Research grade cigarette | 0.49± 0.71 | 1.91± 0.52 | 0.58± 0.42 | 0.80± 0.48 |
| 1R6F | Research grade cigarette | 101.60± 3.40 | 117.90± 29.10 | 110.00± 5.00 | 98.30± 6.70 |
| | Black (Djarum) | 5.02± 3.05 | 4.02± 1.09 | 32.42± 29.78 | 47.86± 45.45 |
| | Special (Djarum) | 329.50± 22.50 | 400.00± 0.00 | 381.50± 18.50 | 397.00± 3.00 |
| Black and Mild | Casino (Black and Mild) | 150.50± 29.50 | 175.50± 5.50 | 117.50± 17.50 | 77.00± 3.00 |
| | Crème (Black and Mild) | 400.00± 0.00 | 400.00± 0.00 | 400.00± 0.00 | 396.00± 0.00 |
| | Jazz (Black and Mild) | 114.00± 20.00 | 160.00± 12.00 | 125.50± 3.50 | 154.00± 16.00 |
| | Regular (Black and Mild) | 180.00± 72.00 | 195.50± 48.50 | 158.00± 1.00 | 304.50± 75.50 |
| Tobacco | Emerald (White Owl) | 55.00± 16.00 | 114.00± 31.00 | 104.50± 20.50 | 151.50± 26.50 |
| | Green Sweets (White Owl) | 64.60± 23.00 | 143.50± 15.50 | 110.45± 22.55 | 80.90± 37.10 |
| | Red Sweet (GAME) | 72.00± 8.00 | 144.50± 51.50 | 89.00± 29.00 | 84.50± 14.50 |
| | Sweet Aromatic (GAME) | 14.90± 3.4 | 28.40± 0.40 | 23.45± 2.55 | 38.05± 0.35 |
| | Tobacco (Swisher Sweets) | 300.50± 61.50 | 248.50± 75.50 | 390.00± 10.00 | 363.00± 37.00 |
| Menthol | Artic Ices (Swisher Sweets) | 398.00± 0.00 | 275.00± 65.00 | 213.50± 65.50 | 371.00± 3.00 |
| | Mint Fusion (Dutch Masters) | 227.00± 40.00 | 216.00± 22.00 | 209.00± 39.00 | 154.05± 69.95 |
| Fruit | Blueberry (Swisher Sweets) | 114.45±21.55 | 86.65± 18.35 | 70.45± 9.95 | 108.50± 13.50 |
| | Strawberry (White Owl) | 146.85± 129.15 | 17.73± 17.06 | 116.90± 92.10 | 56.30± 15.60 |
| | Cherry (Jackpot) | 299.00± 14.00 | 286.00± 114.00 | 350.00± 18.00 | 281.00± 32.00 |
| | Grape (GAME) | 153.00± 16.00 | 110.50± 2.50 | 134.00± 4.00 | 144.50± 27.50 |
| | Grape (Swisher Sweets) | 260.50± 38.50 | 373.00± 27.00 | 379.00± 21.00 | 278.00± 41.00 |
| | Grape (White Owl) | 161.50± 26.50 | 108.00± 16.00 | 73.50± 17.50 | 154.00± 34.00 |
| | Grape (Zig Zag) | 55.70± 21.90 | 125.00± 21.00 | 216.50± 91.50 | 132.35± 62.65 |
| | Mango (Swisher Sweets) | 354.50± 27.50 | 368.00± 26.00 | 242.50± 156.50 | 310.50± 89.50 |
| | Peach (Swisher Sweets | 79.40± 15.60 | 106.50± 2.50 | 123.05± 23.95 | 128.05± 40.95 |
| | Peach (White Owl) | 92.20± 19.80 | 100.65± 6.35 | 108.85± 19.15 | 161.00± 6.00 |
| | Pineapple (GAME) | 73.55± 2.05 | 50.00± 1.70 | 52.20± 2.30 | 72.20± 14.00 |
| | Pineapple (White Owl) | 137.50± 4.50 | 117.50± 8.50 | 124.00± 15.00 | 75.15± 6.85 |
| | Tropical Fusion (Swisher Sweets) | 140.00± 4.00 | 129.00± 0.00 | 202.00± 0.00 | 197.00± 10.00 |
| | Tropical Fusion (Swisher Sweets) | 140.00± 4.00 | 129.00± 0.00 | 202.00± 0.00 | 197.00± 10.00 |
| | Berry Fusion (Dutch Masters) | 24.50± 5.50 | 89.00± 2.00 | 112.50± 35.50 | 75.50± 8.50 |
| Candy | Brownie (GAME) | 298.00± 59.00 | 292.00± 14.00 | 210.00± 53.00 | 290.00± 63.00 |
| | Carmel Peach (Swisher Sweets) | 286.00± 114.00 | 400.00± 0.00 | 316.50± 44.50 | 227.50± 14.50 |
| | Chocolate (Swisher Sweets) | 216.00± 126.00 | 297.00± 79.00 | 382.50± 17.50 | 219.00± 39.00 |
| | Honey (GAME) | 261.50± 24.50 | 266.50± 14.50 | 258.00± 20.00 | 320.00± 47.00 |
| Drinks | Java Fusion (Dutch Masters) | 303.50± 96.50 | 380.50± 19.50 | 268.50± 12.50 | 392.00± 6.00 |
| | Honey Bourbon (White Owl) | 371.50± 28.50 | 253.00± 11.00 | 397.50± 2.50 | 359.00± 41.00 |
| | Sangria (White Owl) | 273.00± 9.00 | 395.50± 4.50 | 398.00± 2.00 | 394.50± 5.50 |
| | Spiked Lemonade (White Owl) | 354.50± 45.40 | 365.00± 25.00 | 392.00± 8.00 | 400.00± 0.00 |
| | Swerve (Swisher Sweets) | 333.33± 66.66 | 221.50± 61.50 | 394.50± 5.50 | 400.00± 0.00 |
| | Wine (Black and Mild) | 316.50± 83.50 | 276.00± 123.50 | 386.00± 14.00 | 390.50± 9.50 |
| Spices | Cream Vanilla (Swisher Sweets) | 317.00± 6.00 | 307.00± 5.00 | 216.50± 61.50 | 234.00± 11.00 |
| | Vanilla (White Owl) | 151.00± 11.00 | 141.50± 15.50 | 246.00± 33.00 | 323.50± 70.50 |

The maximum concentration of particulate matter (sizes 1.0, 2.5, 4.0 and 10.0 μm) for each filter size over the course of a one minute test are reported in the table. Values are given in mg/m$^3$. Each cigarillo or cigar tested in listed under their respective categories, their name and brand is reported as it appears on the wrapper. Data are reported as mean ± SEM.

B

A

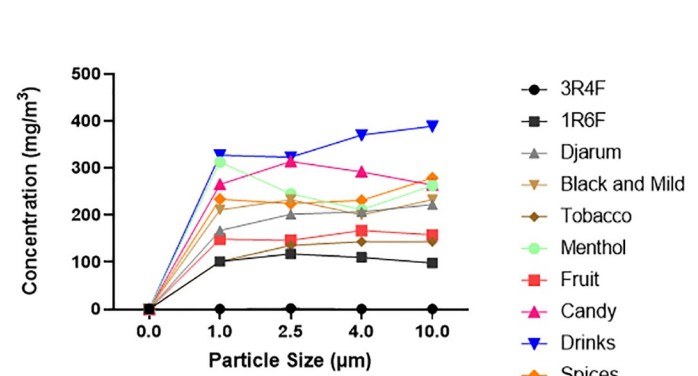
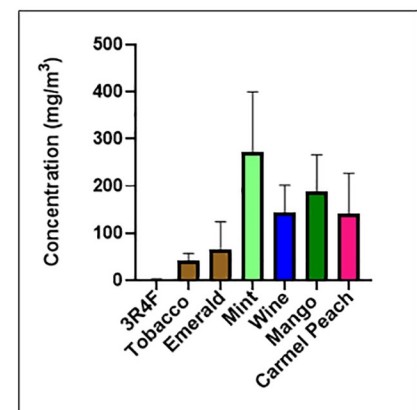

**Fig 7. Particulate matter from flavored cigars.** A) Maximum concentrations of particulate matter in 1.0 filter from representative cigars. Cigarillos and a 3R4F cigarette were each burned two separate times for one minute. Each time when the cigar was burned, the concentration of particles that were 1.0 μm or smaller was obtained and the maximum in that minute was plotted. Data are shown as mean ± SEM. B) The maximum concentration of particulate matter from key representative cigars. Measurements were taken for one minute for each filter size (1.0, 2.5, 4.0, and 10.0 μm). The average maximum concentration for each particle size within the minute of the burning of cigars was plotted.

Most notably, between methods #1 and #2, it is easy to see that each method gives significantly different results even when testing the same cigarillo. Within method #1, many of the cigarillos burned down faster than others from the same brand and flavor, which might use multiple cigars during one run of measurement that might cause batch-batch variations. For example, wine flavors from methods one and two one can again see a wide variation, where PT Rillos is found to be significantly higher than Dutch Masters using method #2, but method #1 shows an opposite trend.Currently, regulations on cigarettes focus on very few ingredients when compared to the total 5,000 chemicals found in them, allowing for variation between brands and batches [27]. This variation is also seen in other tobacco products, such as e-cigarettes [28, 29]. In agreement with our findings, Hamad *et al* recently reported that the flavored cigars/cigarillos (Cheyenne menthol and Swisher sweets original and cherry) have more potentially harmful constituents (HPHCs) including volatile organic hydrocarbons, increased tobacco mass, and total particulate matter (TPM) compared with conventional research reference cigarettes (3R4F) [4]. Hence, HPHCs and particulate matter (PM) including ROS released by cigars would contribute to cellular/tissue injury in the lungs. Cigar smoke produces fine particles altering iron homeostasis, which would easily deposit in the lungs and leading to damage of nearby lung tissue [18]. Our data showed various ranges of particles from flavored cigars (from 1.0 μm to 4.0 μm with higher mass concentration) vs. research-grade research and filtered cigars. In flavored cigars, the particle distribution/concentration was significantly higher in compared to other filtered cigars or research cigarettes, suggesting that composition of flavored cigars generate broader particles with higher concentrations which might be even more harmful than cigarette smoke [30–32]. Further studies are required to determine the grading of various commercially available cigars for their HPHCs, volatile organic compounds (VOCs), and TPM, and their products chemical analyses to ascertain the toxicity in the realm of cellular toxicity especially with flavored cigars/cigarillos and flavoring chemicals used.

In a recent study, it has been shown that increased concentration of the cigar extract led to increased cell death [1, 33]. Although our study found that cigarettes had a higher percentage of cell death compared to most cigarillos or cigars, other studies have noted little cigars to be more toxic when compared to cigarettes [1, 33]. This could be due to the increased amount of

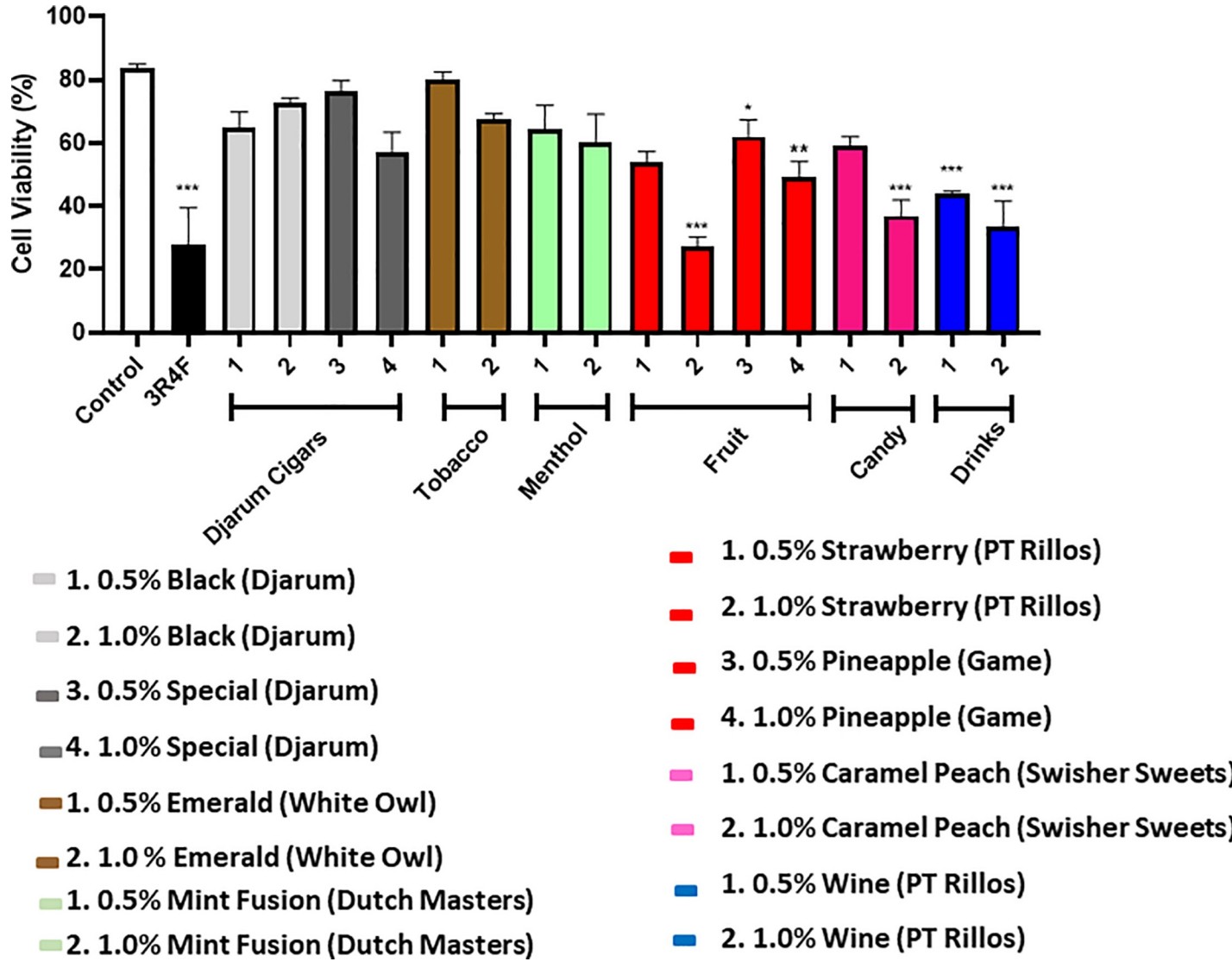

**Fig 8. CSE treatment induces cell toxicity in human bronchial epithelium.** Beas 2B cells were treated with various flavored and unflavored CSE for 24 hours. Each cigarillo treatment was done in 2 concentrations, 0.5% and 1.0%, these are depicted respectively for each group. Flavor is listed with brand in parenthesis. A 3R4F reference cigarette was used for comparison and untreated cells under the same conditions were used as a control. Cell viability was measured using Nexcelom's Cellometer and AOPI dye. Viability percentages are shown as mean ± SEM, and significance was determined by one-way ANOVA. * $p < 0.05$, ** $p < 0.01$, and *** $p < 0.001$ as compared with controls.

chemicals that are found/released in little cigars as compared to cigarettes [33], or the differential interactions of various chemicals during combustion and cellular components. When compring at other tobacco products like e-cigarettes, cell damage still occurs even with the absence of nicotine [34, 35].

Other studies have shown, in particular, the flavorings cinnamaldehyde, O-vanillin, and pentanedione are significantly more cytotoxic on human cells than other flavoring chemicals [36]. Cinnamaldehyde and other aldehyde compounds are frequently used as flavorings, although the FDA and Flavors Extract Manufacturers Association (FEMA) both agree they have adverse effects on human health [37]. Grape and alcohol were common flavors for cigarillos and little cigars. The grape flavoring in food is often attributed to the compound methyl

anthranilate, and alcohol flavors come from ethyl-decadienoate. These compounds are considered safe in food; however, there is little information on the combustion and inhalation toxicology effects of these chemicals [38]. Our data demonstrated that higher cytotoxic responses with fruits/candy and drinks/beverages/alcohols flavored cigars compared to tobacco flavor, which points the harmful possibility of the flavored chemicals. More detailed studies of those flavored chemicals, such as methyl anthranilate or ethyl-decadienoate, are needed to further understand the mechanisms.

Due to the high variability of each manufacturer's/company's flavorings (including batch to batch variations), future research should be directed with cellular methods and mouse models to take a further step in studying the respiratory toxicity [35]. In addition, it would also be beneficial to determine which chemical flavoring (in flavors) is the most deleterious. This has already been conducted in flavors present in e-liquids by studying barrier dysfunction and cellular toxicity in a cellular system [39]. Once more information on the chemical make-up of flavorings and cigars are obtained, the regulatory agencies should increase regulations and/or ban these flavored cigars. Our data showing differential responses due to the different flavors, manufacturers may limit the use of the chemical ingredients in the most toxic flavors (subject to products analysis), or remove flavors all together from cigar products.

In conclusion, despite the high brand-to-brand variability between flavored cigars or cigarillos, those with fruit, candy, and drink flavors tend to show more deleterious effects in our assays. Further, it is difficult to ascertain the consistent toxicity data due to the differences in chemical composition within a specific flavor of the cigar and cigarillo. Flavored cigars produced maximum particles/particulates compared with research cigarettes. The cellular assays resulted in variable cytotoxic responses with fruits/candy and drinks/beverages being more cytotoxic. Due to the irregular composition of flavors and flavoring chemicals, it is important to regulate flavored little cigars/cigarillos. Future research may be directed to determine the cellular methods as well as *in vivo* models to fully determine the flavoring toxicity in flavored little cigars/cigarillos.

## Supporting information

**S1 Table. A complete list of all little cigars and cigarillos tested based on flavor category and sub-category in this work.**
(XLSX)

## Author Contributions

**Conceptualization:** Irfan Rahman.

**Data curation:** Gina R. Lawyer, Thomas Lamb, Qixin Wang, Thivanka Muthumalage.

**Formal analysis:** Thomas Lamb, Qixin Wang, Thivanka Muthumalage.

**Funding acquisition:** Irfan Rahman.

**Investigation:** Gina R. Lawyer, Monica Jackson, Melanie Prinz, Irfan Rahman.

**Methodology:** Gina R. Lawyer.

**Project administration:** Irfan Rahman.

**Supervision:** Thivanka Muthumalage, Irfan Rahman.

**Writing – original draft:** Gina R. Lawyer, Monica Jackson, Irfan Rahman.

**Writing – review & editing:** Gina R. Lawyer, Monica Jackson, Thivanka Muthumalage, Irfan Rahman.

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
