## [Decision Letter · Decision Letter 0]

29 Aug 2019

PONE-D-19-21934

Phylogenic classification of flavors in cigarillos and little cigars and their variable cellular and acellular oxidative and cytotoxic responses

PLOS ONE

Dear Dr. Rahman,

Thank you for submitting your manuscript to PLOS ONE. After careful consideration, we feel that it has merit but does not fully meet PLOS ONE’s publication criteria as it currently stands. Therefore, we invite you to submit a revised version of the manuscript that addresses the points raised during the review process.

We would appreciate receiving your revised manuscript by October 5th, 2019. To enhance the reproducibility of your results, we recommend that if applicable you deposit your laboratory protocols in protocols.io, where a protocol can be assigned its own identifier (DOI) such that it can be cited independently in the future. For instructions see: http://journals.plos.org/plosone/s/submission-guidelines#loc-laboratory-protocols

We look forward to receiving your revised manuscript.

Kind regards,

M. Firoze Khan, Ph.D.

Academic Editor

PLOS ONE

Journal Requirements:

2. Thank you for submitting this work to PLOS ONE. Please supply as a new Supplementary Information document a complete list of all the little cigars and cigarillos tested in this work. The document should include the brand name of the little cigar or cigarillo as well as the primary flavour and flavour subtype.

This work was supported in part by a National Institutes of Health (NIH) Grants, NIH 1R01HL135613, NIH 1R01HL085613, and the Food and Drug Administration (FDA) Center for Tobacco Products (CTP). Also in part was supported by the National Cancer Institute of the National Institutes of Health (NIH) and the Food and Drug Administration (FDA) Center for Tobacco Products under Award Number U54CA228110.

Reviewers' comments:

Reviewer's Responses to Questions

**Comments to the Author**

1. Is the manuscript technically sound, and do the data support the conclusions?

Reviewer #1: Yes

Reviewer #2: Yes

2. Has the statistical analysis been performed appropriately and rigorously? 

Reviewer #1: Yes

Reviewer #2: Yes

3. Have the authors made all data underlying the findings in their manuscript fully available?

Reviewer #1: Yes

Reviewer #2: Yes

4. Is the manuscript presented in an intelligible fashion and written in standard English?

Reviewer #1: Yes

Reviewer #2: Yes

5. Review Comments to the Author

Reviewer #1: The manuscript by Jackson et al presented a novel aspect of phylogenic classification of flavors in cigarillos and little cigars and their variable cellular and acellular oxidative and cytotoxic responses. Flavored and flavorings in tobacco products are emerging. This is the first ever classification of flavors based on phylogeny.

The authors have also studied the oxidative stress and cellular responses by different flavors based on their phylogenic classifications. Most importantly, they showed different flavored cigars/cigarillos had variability in tested parameters between flavors as well as brands of the same flavor with their particle size and cellular toxicity. They found that fruits/candy and drinks/beverages were more toxic which are interesting.

Overall, this is the first study which attempted to classify flavors based on phylogeny. The authors must be commended for their efforts. This classification may be useful for other tobacco products including e-cigarette flavors. Experimental design with data rigor and reproducibility are paramount to this study with solid data presentation, innovative design, and outstanding nature of studies.

Cellular and acellular data presented which are directly correlated with the classified flavors. Particle sizes from flavored cigars are presented using a standard approach. The manuscript is well-written. However, the manuscript needs some careful data organization and analyses.

Major Comments:

1. Table should contain more flavors based on the classified tree of flavors.

2. Particulate matter concentrations from different flavored cigars based on classified flavor tree should be presented in detail in table 1.

3. Figure 2: should have the level of significance by more repeats and careful consideration of flavors. Figures 2B and C should contain statistical analyses. Fruits and beverages drinks should be highlighted with their data in this figure and other figures based on their conclusion.

4. Fig 3B,C should contain statistical analyses.

5. Fig 4: should contain control cigar/cigarillos like in other data figures.

6. Fig 5: more flavors should be studied based on classified flavors.

Reviewer #2: The authors report phylogenetic classification of different flavors used in commercial cigars and cigarillos and their results on cellular and acellular oxidative and cytotoxic responses to these flavors in lung cell models. This is a novel study characterizing flavors using a novel strategy based on phylogenetic tree comparisons. Such analysis can be a prototype for testing other flavors like those for e-cigs. The study is based on logical measurements targeting ROS and cytotoxicity in relation to particulate size distribution. These data are striking in the sense that the flavors showed differential effects. This is the first report providing insights into and demonstrating health effects of flavor-infused cigars and cigarillos and may serve as a useful dataset for regulation of flavoring compounds.

Overall it’s a well written manuscript but attention is drawn to the following aspects:

1. Table 1. More flavors need to be added for their characterization as listed by the authors.

2. Figure 1. The phylogenetic tree needs more clarity and bigger font size.

3. Figures 2 and 3. Need detailed information on vendors and brands of cigars and their ROS production.

4. Figure 4. Information on production of ROS by different portions of cigars may be meaningful.

5. The authors should provide particulate matter data from other flavored cigars as listed in figures 1 and 6.

6. Figure 6: How do Cytotoxicity data correlate with ROS data and particulate data.

7. The authors may pick key flavors and correlate them with ROS generation potential.

Additionally, the authors should fix the following:

1. Replace the term ‘phylogenic’ with the correct term which is ‘phylogenetic’

2. Format the References list per the PLOSone format by cleaning up the extra information.

6. PLOS authors have the option to publish the peer review history of their article (what does this mean?). If published, this will include your full peer review and any attached files.

Reviewer #1: No

Reviewer #2: No

---

## [Author Response · Author response to Decision Letter 0]

22 Oct 2019

Manuscript ID: PONE-D-19-21934

Title: Phylogenic classification of flavors in cigarillos and little cigars and their variable cellular and acellular oxidative and cytotoxic responses

Authors: Gina R Lawyer*, Monica Jackson*, Melanie Prinz, Thomas Lamb, Qixin, Wang, Thivanka Muthumalage, and Irfan Rahman

Overall Response to Reviewers:

We are very grateful for the time the editors and reviewers took to review this manuscript and provided their valuable comments/suggestions. We made sure to review our manuscript thoroughly and answer each question or suggestion that is posed. 

Comments and Responses: 

Reviewer #1: 

The manuscript by Jackson et al presented a novel aspect of phylogenic classification of flavors in cigarillos and little cigars and their variable cellular and acellular oxidative and cytotoxic responses. Flavored and flavorings in tobacco products are emerging. This is the first ever classification of flavors based on phylogeny.

The authors have also studied the oxidative stress and cellular responses by different flavors based on their phylogenic classifications. Most importantly, they showed different flavored cigars/cigarillos had variability in tested parameters between flavors as well as brands of the same flavor with their particle size and cellular toxicity. They found that fruits/candy and drinks/beverages were more toxic which are interesting.

Overall, this is the first study which attempted to classify flavors based on phylogeny. The authors must be commended for their efforts. This classification may be useful for other tobacco products including e-cigarette flavors. Experimental design with data rigor and reproducibility are paramount to this study with solid data presentation, innovative design, and outstanding nature of studies.

Cellular and acellular data presented which are directly correlated with the classified flavors. Particle sizes from flavored cigars are presented using a standard approach. The manuscript is well-written. However, the manuscript needs some careful data organization and analyses.

Response: We are extremely thankful for the kind comments and recognition. We have made sure our best effort to review the data organization and analysis to improve this manuscript.

Major Comments: Table should contain more flavors based on the classified tree of flavors

Response 1: We agree with the review’s comments, we also think more flavors based on our classification are needed and it’s really necessary for quantification and validation. Please check Figures 3 and 5, and Table 1, as suggested, we have ran several more cigars for ROS, and for particulate matter analyses in the revised manuscript.

Major Comments 2: Particulate matter concentrations from different flavored cigars based on classified flavor tree should be presented in detail in table 1.

Response 2. We are thankful for this kind suggestion regarding better understanding of physical properties of different flavored cigars. As showing in our revised Table 1, different flavored cigars are added, and physical particular matter concentrations are shown in 4 different diameters with units of mg/m3 (see Table 1 and Figure 5).

Major Comments 3: Figure 2: should have the level of significance by more repeats and careful consideration of flavors. Figures 2B and C should contain statistical analyses. Fruits and beverages drinks should be highlighted with their data in this figure and other figures based on their conclusion.

Response 3: We agree with the reviewer’s comments and concerns. We have carefully analyzed more acellular ROS using different flavored cigars, and analyze the statistical significance accordingly. As we mentioned in our discussion, due to the batch-batch variation of the commercial available cigars, part of results are still scattered. However, most of the cigars showed significant harmful effects even compared to research cigarette smoke. 

We are really thankful for the suggestion of highlight the fruits/drinks flavored cigars. We have revised it accordingly in our results and discussion in the main text of the revised manuscript.

Major Comments 4: Fig 3B, C should contain statistical analyses.

Response 4: We thank the reviewer for their suggestion. We have applied statistical analysis accordingly to Figures 3b and 3c. 

Major Comments 5: Fig 4: should contain control cigar/cigarillos like in other data figures.

Response 5: We thank the reviewer for their suggestion. We have added a control of air to figure 4. 

Major Comment 6: Fig 5: more flavors should be studied based on classified flavors.

Response 6: We thank the reviewer for their suggestion. More flavors have been tested for their particulate matter and acellular ROS. In Fig 3, more flavored cigars have been tested in ROS, several flavored cigars have been tested in this study. In table 1, several flavored cigars have been characterized with PM distribution and concentration; various flavored cigars have been measured. 

Reviewer #2:

The authors report phylogenetic classification of different flavors used in commercial cigars and cigarillos and their results on cellular and acellular oxidative and cytotoxic responses to these flavors in lung cell models. This is a novel study characterizing flavors using a novel strategy based on phylogenetic tree comparisons. Such analysis can be a prototype for testing other flavors like those for e-cigs. The study is based on logical measurements targeting ROS and cytotoxicity in relation to particulate size distribution. These data are striking in the sense that the flavors showed differential effects. This is the first report providing insights into and demonstrating health effects of flavor-infused cigars and cigarillos and may serve as a useful dataset for regulation of flavoring compounds.

Response: We are extremely thankful for reviewer’s 2 encouraging comments and suggestions. We made sure to thoroughly address each suggestion or question.

Major Comment 1: Table 1. More flavors need to be added for their characterization as listed by the authors.

Response 1: We agree with the reviewer’s comments. We have addressed this comment previously. Please refer to the earlier response (reviewer #1, comment #1).

Major Comments 2: Figure 1. The phylogenetic tree needs more clarity and bigger font size.

Response 2: As suggested, we have increased the size of Figure 1 and the font within the figure so that it is clearer for readers.

Major Comments 3: Figures 2 and 3. Need detailed information on vendors and brands of cigars and their ROS production.

Response 3: We agree with the reviewer’s comments. We have the details that described the information of each cigar used with their brand, flavor based on packaging, and which category they have been included in the revised manuscript.

Major Comment 4: Figure 4. Information on production of ROS by different portions of cigars may be meaningful.

Response 4: We agree with the review’s comments. We have cut the cigar in different portion and measured the ROS generation. As showing in Figure 4, there is no significant difference among different portions. 

Major Comment 5. The authors should provide particulate matter data from other flavored cigars as listed in figures 1 and 6.

Response 5: As suggested, more cigars have been tested and included into Figure 5 and table 1. We have addressed the similar comments (see above); please check the previous responses (reviewer #1, comments #6)

Major Comment 6: Figure 6: How do Cytotoxicity data correlate with ROS data and particulate data.

Response 6: We agree and understand reviewer’s comments and concerns. Generation of particulate matter affects the cellular uptake of chemicals or particles from cigar combustion, and it does participate in the cellular toxicity responses, as well as acellular ROS produced by cigars. However, there are more factors affects the cigars, such as the chemical compounds and deposition of cigar particles, all have cytotoxic influence. Though, we cannot fully connect the ROS and particulate matter data to our cell viability results, more detailed studies, such as cell toxicity test with different flavored chemicals, are in progress to figure out what is the key points caused cytotoxicity.

Major Comment 7: The authors may pick key flavors and correlate them with ROS generation potential.

Response 7: we thank for the reviewer’s comments/suggestions. We have picked up the key flavors that represent each flavoring category, as well as more cigars were used to test the ROS and particulate matter. Because of the cigars from different brands and even the different batch showed variations, there is no significantly specific trend across flavor and ROS. However, all the flavoring cigars showed higher ROS generation than research cigarette smoke.

Additional notes from Reviewer #2: Additionally, the authors should fix the following: 1-Replace the term ‘phylogenic’ with the correct term which is ‘phylogenetic’. 2-Format the References list per the PLOSone format by cleaning up the extra information.

Additional Notes Response: We thank the reviewer for his/her valuable suggestion, we have corrected and replaced accordingly.

---

## [Editor Report · Decision Letter 1]

5 Nov 2019

PONE-D-19-21934R1

Phylogenic classification of flavors in cigarillos and little cigars and their variable cellular and acellular oxidative and cytotoxic responses

PLOS ONE

Dear Dr. Rahman,

Thank you for submitting your manuscript to PLOS ONE. After careful consideration, we feel that it has merit but does not fully meet PLOS ONE’s publication criteria as it currently stands. Therefore, we invite you to submit a revised version of the manuscript that addresses the points raised during the review process.

The Academic Editor has reviewed your revised manuscript and notes that the revisions have appropriately addressed the reviewers' concerns. However, the PLOS ONE staff editors have identified one aspect of the study methodology and some minor concerns with the text that we think should be addressed before we proceed. Please note that we have discussed these concerns with the Academic Editor, who agrees that these minor issues should be fixed. They are as follows: 

1) We note the use of the T-REX web server and iTOL tool to develop a phylogenetic tree of cigarillo and little cigar flavorings. We are a concerned about the development of a “phylogenetic tree” in this setting, as phylogenetic trees are generally used to show evolutionary relationships, which is not what is reported in this paper. We would propose that this is instead referred to as a “classification”. This would involve several edits to the text, as follows:

- Title: Change to “Classification of flavors in cigarillos and little cigars and their variable cellular and acellular oxidative and cytotoxic responses”

- Abstract: Change “A new phylogenic classification system…” to “A new classification system…”

- Introduction, p4: Delete the following sentences: “In biological classification, organisms are put into categories based on shared traits [20]. The closer organisms are in the tree, the more related they are to each other [20]. Although the amount of different cigar flavors is vast, the system used to classify these flavors must be able to convey a large amount of information clearly.”

- Results p9: Change “To understand how the flavorings in cigars can be grouped a phylogenic tree was created” to “To understand how the flavorings in cigars can be grouped a new classification system was created”

- Discussion p12: Change “Classification of flavoured cigars/cigarillos in a phylogenic tree should provide a convenient nomenclature/vocabulary in the scientific community as there was no scientific classification of cigar flavorings until the present” to “Classification of flavoured cigars/cigarillos should provide a convenient nomenclature/vocabulary in the scientific community as there was no scientific classification of cigar flavorings until the present”

- Discussion p12: Change “The tree created allows for the categorization of flavors…” to “The classification system created allows for the categorization of flavors…”

The tree presented in figure 1 should be changed to a different visual representation, for example a table showing to which flavour category and sub-category each flavour belongs. The legend to figure 1, as well as the Methods, should then be updated to remove the references to phylogenetic trees.

2) We would like to suggest some changes to the text for clarity:

- Abstract: We note that none of the flavouring compounds tested in this work was as cytotoxic as cigarette smoke (in the Results, the authors state “Almost all cigars tested had a reduced in cell viability compared with the control, but were not more cytotoxic than smoke cigarette”). We think this finding should be added to the Abstract, for example “A differential cytotoxic response was observed with cigarillo smoke extract treatments: “fruits/candy” and “drinks” were the most toxic, but were not more cytotoxic than smoke from cigarettes.”

- Discussion p12: The phrase “tobacco companies are creating more non-descriptive names to circumvent the regulation” may not be fully supported by the reference provided as the reference is from 2016 and that statement suggests that this practice is ongoing. We would suggest that this statement be removed; it should be sufficient to note that the introduction of additional flavourings will make comparison between categories more difficult.

3) As you may know, PLOS ONE does not copyedit text before publication. Thus, please ensure that the text is copyedited for grammar and usage before the manuscript is resubmitted.

We would appreciate receiving your revised manuscript by Dec 20 2019 11:59PM. To enhance the reproducibility of your results, we recommend that if applicable you deposit your laboratory protocols in protocols.io, where a protocol can be assigned its own identifier (DOI) such that it can be cited independently in the future. For instructions see: http://journals.plos.org/plosone/s/submission-guidelines#loc-laboratory-protocols

We look forward to receiving your revised manuscript.

Kind regards,

Emily Chenette, Staff Editor, PLOS ONE

On behalf of

M. Firoze Khan, Ph.D.

Academic Editor

PLOS ONE

---

## [Author Response · Author response to Decision Letter 1]

10 Nov 2019

Manuscript ID: PONE-D-19-21934R2

Title: Classification of flavors in cigarillos and little cigars and their variable cellular and acellular oxidative and cytotoxic responses

Authors: Gina R Lawyer*, Monica Jackson*, Melanie Prinz, Thomas Lamb, Qixin, Wang, Thivanka Muthumalage, and Irfan Rahman

Academic Editor Comments:

The Academic Editor has reviewed your revised manuscript and notes that the revisions have appropriately addressed the reviewers' concerns. However, the PLOS ONE staff editors have identified one aspect of the study methodology and some minor concerns with the text that we think should be addressed before we proceed. Please note that we have discussed these concerns with the Academic Editor, who agrees that these minor issues should be fixed. They are as follows: 

Specific Comments:

1) We note the use of the T-REX web server and iTOL tool to develop a phylogenetic tree of cigarillo and little cigar flavorings. We are a concerned about the development of a “phylogenetic tree” in this setting, as phylogenetic trees are generally used to show evolutionary relationships, which is not what is reported in this paper. We would propose that this is instead referred to as a “classification”. This would involve several edits to the text, as follows:

- Title: Change to “Classification of flavors in cigarillos and little cigars and their variable cellular and acellular oxidative and cytotoxic responses”

Response: We have changed the title as suggested.

Comment:

- Abstract: Change “A new phylogenic classification system…” to “A new classification system…”

Response: This has been changed.

Comment:

- Introduction, p4: Delete the following sentences: “In biological classification, organisms are put into categories based on shared traits [20]. The closer organisms are in the tree, the more related they are to each other [20]. Although the amount of different cigar flavors is vast, the system used to classify these flavors must be able to convey a large amount of information clearly.”

Response: This has been amended.

Comment:

- Results p9: Change “To understand how the flavorings in cigars can be grouped a phylogenic tree was created” to “To understand how the flavorings in cigars can be grouped a new classification system was created”

Response: This has been changed.

Comment:

- Discussion p12: Change “Classification of flavoured cigars/cigarillos in a phylogenic tree should provide a convenient nomenclature/vocabulary in the scientific community as there was no scientific classification of cigar flavorings until the present” to “Classification of flavoured cigars/cigarillos should provide a convenient nomenclature/vocabulary in the scientific community as there was no scientific classification of cigar flavorings until the present”

Response: This has been changed.

Comment:

- Discussion p12: Change “The tree created allows for the categorization of flavors…” to “The classification system created allows for the categorization of flavors…”

 Response: This has been changed.

Comment:

The tree presented in figure 1 should be changed to a different visual representation, for example a table showing to which flavour category and sub-category each flavour belongs. The legend to figure 1, as well as the Methods, should then be updated to remove the references to phylogenetic trees.

 Response: We have changed the Figure 1, and legend is amended as suggested.

Comment:

2) We would like to suggest some changes to the text for clarity:

- Abstract: We note that none of the flavouring compounds tested in this work was as cytotoxic as cigarette smoke (in the Results, the authors state “Almost all cigars tested had a reduced in cell viability compared with the control, but were not more cytotoxic than smoke cigarette”). We think this finding should be added to the Abstract, for example “A differential cytotoxic response was observed with cigarillo smoke extract treatments: “fruits/candy” and “drinks” were the most toxic, but were not more cytotoxic than smoke from cigarettes.”

Response: This has been changed as suggested.

Comment:

- Discussion p12: The phrase “tobacco companies are creating more non-descriptive names to circumvent the regulation” may not be fully supported by the reference provided as the reference is from 2016 and that statement suggests that this practice is ongoing. We would suggest that this statement be removed; it should be sufficient to note that the introduction of additional flavourings will make comparison between categories more difficult.

 Response: This has been changed as suggested

3) As you may know, PLOS ONE does not copyedit text before publication. Thus, please ensure that the text is copyedited for grammar and usage before the manuscript is resubmitted.

Response: we have now copyedited for grammar and usage in the revised manuscript.

---

## [Editor Report · Decision Letter 2]

20 Nov 2019

Classification of flavors in cigarillos and little cigars and their variable cellular and acellular oxidative and cytotoxic responses

PONE-D-19-21934R2

Dear Dr. Rahman,

We are pleased to inform you that your manuscript has been judged scientifically suitable for publication and will be formally accepted for publication once it complies with all outstanding technical requirements.

With kind regards,

M. Firoze Khan, Ph.D.

Academic Editor

PLOS ONE

Journal Requirements:

Thank you for submitting your revised manuscript to PLOS ONE. As you prepare your manuscript for publication, we ask that you please incorporate the following edits to the text:

- Abstract - The phrase ‘*have a shared vocabulary within the scientific community’* appears to suggest that the community has consulted on and accepted the classification system reported here. As this is the first publication reporting this classification, we suggest that this phrase be removed.

- Abstract – Because the study is performed *in vitro*, please add the word 'potential' to the following sentence: ‘Our study provides insight into understanding the **potential** health effects of flavor-infused cigars/cigarillos and the need for the regulation of those flavoring chemicals in these products’

Page 12 – We suggest that the phrase ‘*Further, this could help not only researchers and public health officials to become more aware of which flavoring chemicals need to be regulated, but also users of cigars or any flavored tobacco product'* be removed, as the chemicals per se were not tested in this work, and there is (as you note) high variability between batches and between flavours in the same category.

Page 15- We would suggest that the phrase *‘Our study highlights the importance of legislation and regulation of all flavored tobacco products. as well as increases the ease of communication in the scientific community*'**be removed as this is not supported by the analyses performed in this work.

Page 15* - *Could you please rephrase the following sentences: *‘In conclusion, due to the high variability from one flavored cigar or cigarillo to the other (including batch to batch variations), fruit, candy and drink flavors show more deleterious effects. It is difficult to ascertain the consistent data due to the differences in chemical composition or flavors of the cigars and cigarillo.’* Currently, it is unclear how variability in batches would cause some flavors to show more deleterious effects, could this please be clarified?
---

## [Editor Report · Acceptance letter]

2 Dec 2019

PONE-D-19-21934R2 

Classification of flavors in cigarillos and little cigars and their variable cellular and acellular oxidative and cytotoxic responses 

Dear Dr. Rahman:

I am pleased to inform you that your manuscript has been deemed suitable for publication in PLOS ONE. Congratulations! Your manuscript is now with our production department. 

With kind regards,

on behalf of

Dr. M. Firoze Khan 

Academic Editor

PLOS ONE